# Transcriptomics Analysis of Heat Stress-Induced Genes in Pepper (*Capsicum annuum* L.) Seedlings

**Fei Wang** †, **Yanxu Yin** †, **Chuying Yu, Ning Li, Sheng Shen, Yabo Liu, Shenghua Gao** , **Chunhai Jiao** * **and Minghua Yao** *

Hubei Key Laboratory of Vegetable Germplasm Enhancement and Genetic Improvement, Cash Crops Research Institute, Hubei Academy of Agricultural Sciences, Wuhan 430070, China; wangfei_raymond@163.com (F.W.); yinyanxu2008@126.com (Y.Y.); yuchuying@126.com (C.Y.); n.li@msn.com (N.L.); shengshen199711@163.com (S.S.); liu13125048135@163.com (Y.L.); gaoshenghua1986@126.com (S.G.)

\* Correspondence: jiaoch@hotmail.com (C.J.); yaomh_2008@126.com (M.Y.); Tel.: +86-27-87380819 (M.Y.)

† These authors contributed equally to this work.

**Abstract:** Pepper (*Capsicum annuum* L.) is one of the most economically important crops worldwide. Heat stress (HS) can significantly reduce pepper yield and quality. However, changes at a molecular level in response to HS and the subsequent recovery are poorly understood. In this study, 17-03 and H1023 were identified as heat-tolerant and heat-sensitive varieties, respectively. Their leaves' transcript abundance was quantified using RNA sequencing to elucidate the effect of HS and subsequent recovery on gene expression. A total of 11,633 differentially expressed genes (DEGs) were identified, and the differential expression of 14 randomly selected DEGs was validated using reverse-transcription polymerase chain reaction. Functional enrichment analysis revealed that the most enriched pathways were metabolic processes under stress and photosynthesis and light harvesting during HS and after recovery from HS. The most significantly enriched pathways of 17-03 and H1023 were the same under HS, but differed during recovery. Furthermore, we identified 38 heat shock factors (Hsps), 17 HS transcription factors (Hsfs) and 38 NAC (NAM, ATAF1/2, and CUC2), and 35 WRKY proteins that were responsive to HS or recovery. These findings facilitate a better understanding of the molecular mechanisms underlying HS and recovery in different pepper genotypes.

**Keywords:** pepper; transcriptomics; heat stress; transcription factor





## 1. Introduction

Pepper (*Capsicum annuum* L.) is an important member of the Solanaceae family and is one of the most important spice and vegetable crops in many countries [1]. It is rich in capsaicin, capsanthin, and vitamins, which can improve appetite and health [2]. Pepper grows well in warm climates but is sensitive to high temperatures, with the suitable temperature range for growth and development being 20–30 °C [3]. When the temperature exceeds 35 °C, the plant will suffer from heat stress (HS) and show symptoms of high temperature injury in the whole growth stage, which will adversely affect the plant morphology, physiological and biochemical metabolic processes, and other aspects [3,4]. With the intensification of the greenhouse effect, global temperatures have risen, impacting the growth and development of crops and presenting a severe challenge for many agricultural regions in the world, and leading to a drastic reduction in economic yields and quality [5]. Therefore, investigating the molecular mechanisms underlying the response of pepper to HS is imperative for developing varieties that are better adapted to more hostile conditions.

HS affects plant cell structure, protein denaturation, and lipid transport, resulting in the destruction of the plasma membrane structure and the death of specific cells or tissues. HS causes plant transpiration water loss, decreased photosynthetic rate, and abnormal metabolism, which affect the growth and development of plants [4]. Photosynthesis is a

very heat-sensitive physiological process and is easily inhibited by HS, affecting almost all photosynthetic processes, including photosystem II, photosystem I, electron transport chain, adenosine triphosphate (ATP) synthesis, and carbon fixation [6,7]. In addition to the decrease in net photosynthetic efficiency and photosystem activity, reactive oxygen species (ROS) accumulate, resulting in the destruction of D1 protein and antenna pigment in serious cases, and thus reducing the ability of plants to absorb and utilize light energy and sequester carbon [8,9]. Additionally, ROS accumulation caused by HS in plants results in oxidative damage to cells. High temperatures cause metabolic imbalances and production of ROS in plants, which aggravate lipid peroxidation and protein denaturation of the cell membrane, thus affecting the structure and function of biofilms; severe cases can lead to cell damage and plant death [10]. High temperatures also greatly effect plant metabolism; for example, most of the genes in the anthocyanin biosynthesis pathway of eggplant are induced and downregulated under high temperatures, resulting in a decrease in anthocyanin accumulation [11]. Many abiotic stresses, including HS, directly or indirectly affect the synthesis, concentration, metabolism, transport, and storage of sugars. As a potential signal molecule, soluble sugars interact with light, nitrogen, and abiotic stresses to regulate plant growth and development [12–14].

Thermal signal perception and transduction are important parts of plant stress resistance, involving a number of signal transduction pathways, including calcium-dependent protein kinases (CDPKs) and mitogen-activated protein kinases (MAPK/MPKs), signal molecules (such as ROS), and plant hormones, which play important roles in various cellular signaling networks, by transmitting extracellular stimuli to generate intracellular responses. Thermal signal perception and transduction actively regulate gene expression and protein function under various stresses and ultimately cause adaptation to environmental stresses [15–19]. For example, CRISPR/Cas9-mediated *SlMAPK3* tomato mutants were more heat-tolerant than wild-type plants, showing less plant wilting and membrane damage, a lower ROS content, higher antioxidant enzyme activities, and higher transcriptional levels [20]. The heat-induced 47 kD MBP-phosphorylated protein SlMPK1 negatively regulates the heat tolerance of tomato by mediating antioxidant protection and redox metabolism; SlSPRH1, a protein homolog rich in serine and proline, is the target protein of SlMPK1 and can be phosphorylated by SlMPK1. Overexpression of SlSPRH1 reduces the heat tolerance and antioxidant capacity of plants and is related to SlSPRH1 phosphorylation. The SlMPK1-SlSPRH1 module negatively regulates the high-temperature signal in the high-temperature response process and cooperates with the antioxidant stress system [21]. Evidence shows that HS is accompanied by a certain degree of oxidative stress, and there is a crosstalk between the signals of heat and oxidative stresses. A study showed that $H_2O_2$ erupts after a short period of time under HS, owing to the activity of NADPH oxidase [22]. This outbreak was related to the induction of HS response genes [23]. $H_2O_2$ or menadione pretreatment can also improve heat tolerance in plants [24]. BZR1, the key regulator of brassinoid (BR) response, regulates the HS response of tomato through RBOH1-dependent ROS signaling; at least in part through the regulation of FER2 and FER3 [25].

Plant heat shock transcription factors (Hsfs) are important regulatory factors of signal transduction, which mediate the transcription of heat shock factors (HSPs) and other HS-induced genes [26]. *HsfA1a* regulates the initial response, and *HsfA1a* and *HsfB1* are constitutively expressed at a steady-state low abundance of mRNA. Under HS, the accumulation of *HsfA2* mRNA and protein is strongly induced, and *HsfA2* becomes the most abundant Hsf, regulating heat tolerance during recovery or after repeated HSs [27–29]. Under non-stress conditions, overexpression of *HsfB1* stimulates the co-activation of *HsfB1*, which promotes the accumulation of HS-related proteins and enhances heat tolerance [30,31]. Hsps are regulated by Hsfs, including *Hsp100/ClpB*, *Hsp90/HtpG*, *Hsp70/DnaK*, *Hsp60/GroEL*, and small *Hsp* (*sHsp*), which are generally considered to be important molecular chaperones for maintaining and/or restoring protein homeostasis, which plays a vital role in plant survival under HS [32,33]. In addition to Hsfs, other large families of transcription factors in plants are also involved in HS responses, such as WRKY, bZIP, MYB, and NAC. As a

downstream negative regulator of the $H_2O_2$-mediated HS response, *CaWRKY27* prevents improper responses during HS and recovery [3]. *CabZIP63*, a member of the bZIP family in pepper, directly or indirectly regulates the expression of *CaWRKY40* at the transcriptional and post-transcriptional levels and forms a positive feedback loop with *CaWRKY40* during the response of pepper to ralstonia solanacearum inoculation or high temperature–high humidity [34]. Overexpression of *SlAN2* induced the upregulation of the expression of several structural genes in the anthocyanin biosynthesis pathway and caused anthocyanin accumulation in tomato, which enhanced the tolerance to HS [35].

In nature, when plants are subjected to HS, their ability to recover is important, as the stronger the ability to recover, the faster the plant can restore their metabolic balance and maintain their normal growth. However, the regulatory molecular mechanisms and networks of pepper have not yet been reported. Therefore, in this study, we performed transcriptome analysis of the heat-tolerant variety 17-03 and the heat-sensitive variety H1023 during HS recovery, to identify candidate genes that had altered transcription levels in the pepper leaves. Collectively, our findings provide a theoretical basis for the cultivation of high-quality, heat-resistant varieties.

## 2. Materials and Methods

### 2.1. Plant Materials and Heat Treatments

Two pepper varieties, heat-tolerant 17-03 and heat-sensitive H1023, were obtained from the Hubei Key Laboratory of Vegetable Germplasm Enhancement and Genetic Improvement, Hubei Academy of Agricultural Sciences, for transcriptome analysis. Seeds were sown in 50-hole trays and grown in a growth chamber under cool white fluorescent lights (approximately 200 μmol/m$^2$/s) at 25 ± 2 °C with a photoperiod of 16 h light/8 h dark and 70–80% relative humidity. At two weeks, seedlings with uniform growth were transplanted to plastic pots (10 × 10 × 10 cm), containing peat, vermiculite, and soil (v/v/v = 1:1:1), until they reached the stage of 6–8 true leaves. For heat treatment, seedlings were well watered and cultivated at 42 °C for 3 d before recovering at 25 °C for 1 d. The control seedlings were placed at 25 °C for 4 d. The lighting conditions and humidity were not changed. The first fully expanded leaf from the top of each plant was sampled from eight seedlings (three replicates) in the control and treatment groups. All samples were immediately frozen in liquid nitrogen and stored at −80 °C for RNA sequencing. The samples were named CK1 (HT1_1-HT1_3) (control group of 17-03), T1 (HT2_1-HT2_3) (heat treatment group of 17-03), M1 (HT3_1-HT3_3) (recovered group of 17-03), CK2 (HS1_1-HS1_3) (control group of H1023), T2 (HS2_1-HS2_3) (heat treatment group of H1023), and M2 (HS3_1-HS3_3) (recovered group of H1023).

### 2.2. Measurement of Relative Electrolyte Leakage and Proline Content

Relative electrolyte leakage, which measures cellular membrane integrity, is frequently used to evaluate plant stress tolerance [36]. The upper third of the fully expanded leaves from treated and non-treated plants of 17-03 and H1023 were excised and used to generate leaf discs (9 mm in diameter). Three replicates were used for each line, with 20 leaf discs per replicate. The leaf discs were placed into 50 mL centrifuge tubes containing 25 mL of distilled deionized water and shaken at 60 rpm for 12 h in the dark at 25 °C. The electrolyte leakage (R1) of the solution was measured using a portable magnetic conductivity meter (DDB-303A, Shanghai, China) at 25 °C. The solutions were boiled for 30 min and then cooled to room temperature. The electrolyte leakage in the boiled solution (R2) was then determined using the same method. The relative electrolyte leakage (%) was calculated as (R1/R2) × 100.

The proline content was measured as described by Ben et al. [37]. Briefly, 200 mg of ground leaf sample was extracted with 2 mL of 3% sulfosalicylic acid in boiling water for 10 min. The samples were cooled and centrifuged at 4000 rpm for 15 min. Then, 1 mL of the supernatant was transferred to a new 15 mL test tube. Then, 1 mL of glacial acetic acid and 1 mL of acid-ninhydrin reagent were added. The mixture was boiled for

30 min and cooled in an ice bath to terminate the reaction. Next, the reaction mixture was partitioned by adding toluene (3 mL). After static delamination, the upper liquid was centrifuged at 4000 rpm for 15 min. The absorbance of the organic phase was measured in a spectrophotometer (Schimadzu, Japan) at a 520 nm wavelength. The proline content of the leaf samples was calculated using a standard curve constructed with known amounts of proline.

### 2.3. RNA Extraction, Library Construction, and Transcriptome Sequencing

According to the manufacturer's instructions, total RNA was extracted from 18 pepper leaf samples using TRIzol reagent (Life Technologies, California, CA, USA). The mRNA with poly (A) in the total RNA was enriched using Oligo (dT) magnetic beads and divided into fragments approximately 300 bp in length using ion interruption. First-strand cDNA was synthesized using a M-MuLV reverse transcriptase system, using these RNA fragments as templates and random hexamer primers, while the second strand cDNA was synthesized using the first-strand cDNA as a template. Subsequently, the cDNA libraries were constructed after polymerase chain reaction (PCR) amplification and selected according to the fragment length of 450 bp. Then, the quality of the cDNA libraries was checked using an Agilent 2100 Bioanalyzer system (Agilent Technologies, Inc., Santa Clara, CA, USA). Based on the Illumina sequencing platform, the qualified libraries were sequenced using a double terminal (paired-end, PE) sequencer (Illumina, Foster, CA, USA).

### 2.4. Read Alignment and Mapping Reads to the Reference Genome

To analyze the sequencing results effectively and accurately, low-quality raw data or connectors in the sequencing data were filtered. Cutadapt was used to remove the 3′ sequencing adapter (https://cutadapt.readthedocs.io/en/stable/) (accessed on 10 March 2021), and reads with an average mass fraction lower than Q20 were removed [38]. The high-quality clean reads from each library were mapped to the pepper reference genome CM334 (https://ftp.solgenomics.net/genomes/Capsicum_annuum/C.annuum_cvCM334/) (accessed on 10 March 2021) using HISAT2 software (http://ccb.jhu.edu/software/hisat2/index.shtml) (accessed on 10 March 2021). The read count value of each gene was mapped using HTSeq as the original expression of the gene [39]. Fragments per kilobase of transcript per million mapped reads (FPKM) was used to standardize the gene expression levels based on Cufflinks software [40].

### 2.5. Functional Enrichment Analysis of Differentially Expressed Genes

Genes with an absolute value of |log2fold change| > 1 and a false discovery rate (FDR) < 0.05 were identified as representing significantly differentially expressed genes (DEGs), using DESeq2 in the four comparisons of CK1_vs_T1, CK1_vs_M1, CK2_vs_T2, and CK2_vs_M2 [41]. To study the putative functions and pathways of the DEGs in the above four comparisons, gene ontology (GO) functional enrichment analysis was conducted using Blast2GO (version 3.0; https://www.blast2go.com/) (accessed on 10 March 2021) [42]. Kyoto Encyclopedia of Genes and Genomes (KEGG) pathway annotation of DEGs was performed using Cytoscape software (version 3.2.0) (https://cytoscape.org/) (accessed on 10 March 2021) with the ClueGO plugin using a hypergeometric test and Benjamini-Hochberg FDR correction (FDR ≤ 0.05) [43]. Transcription factors were predicted using PlantTFDB [44].

### 2.6. Reverse Transcription and Quantitative Reverse-Transcription Polymerase Chain Reaction Analysis

RNA was first treated with DNase I and then reverse-transcribed to cDNA using a HiScript II 1st Strand cDNA Synthesis Kit (+gDNA wiper), according to the manufacturer's instructions (Vazyme, Nanjing, China). Then, the concentration of cDNA was diluted to 100 ng/μL, and the internal reference gene *CaUBI-3* forward and reverse primers were used for PCR to detect whether the reverse transcription was successful [45]. For the quantitative reverse-transcription polymerase chain reaction (qRT-PCR) assay, all primers

were designed using Primer3 (http://primer3.ut.ee/) (accessed on 10 March 2021), and the specificity of the designed fragment was tested using SGN (https://solgenomics.net/) (accessed on 10 March 2021) (Table S8). qRT-PCR was performed using SYBR® Premix Ex Taq™ (Vazyme), with three technical replicates in 10 μL volumes containing 5 μL Fast SYBR™ Green Master Mix (2×), 2 μL cDNA template (100 ng/μL), 0.2 μL of each primer (10 μM), and 2.6 μL ddH$_2$O. The PCR cycling conditions were as follows: 95 °C for 30 s, followed by 40 cycles of 95 °C for 5 s, and finally 60 °C for 20 s. The relative expression level of each gene was calculated using the $2^{-\Delta\Delta CT}$ method [46].

## 3. Results

### 3.1. Phenotypic and Physiological Responses of 17-03 and H1023 under HS

To accurately evaluate the heat resistance of pepper in the seedling stage under HS, 17-03 (heat-tolerant variety) and H1023 (heat-sensitive variety) were treated at 42 °C for 3 d and then recovered for 1 d at 25 °C. Compared with the leaves of seedlings of 17-03 and H1023 that did not undergo HS (Figure 1a,d), the leaves of seedlings 17-03 after heat treatment were slightly bent (Figure 1b), while the leaves of H1023 were severely sagged, and the lower leaves were severely wilted. Additionally, the growth points of the plants were necrotic after treatment at 42 °C for 3 d (Figure 1e). After recovering at 25 °C for 1 d, the down-bent leaves of 17-03 were completely extended (Figure 1c), while those of H1023 only partially recovered and did not stretch out completely. Additionally, the edges of the leaves had different degrees of withering (Figure 1f).

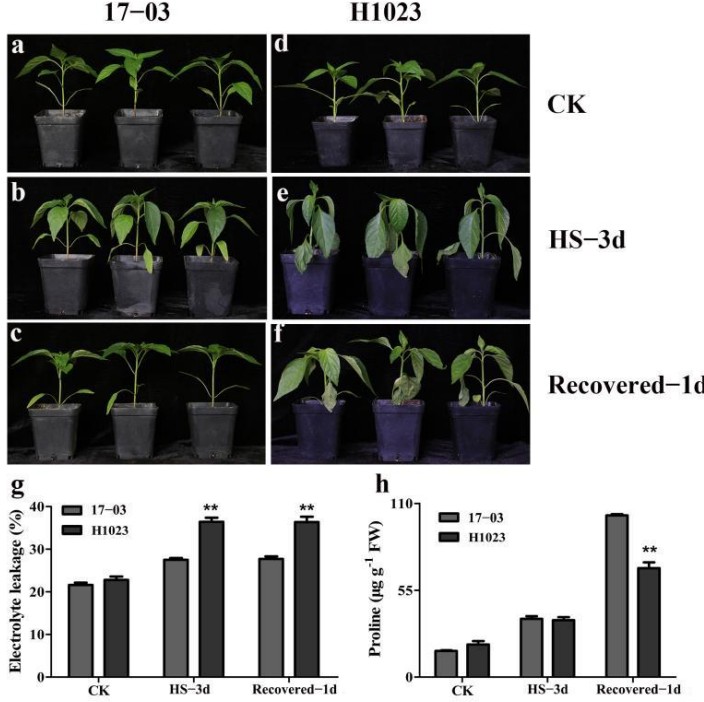

**Figure 1.** Phenotypic and physiological responses of 17-03 and H1023 under heat stress (HS). Seedlings of 17-03 and H1023 in control (**a**,**d**), treated at 42 °C for 3 d (**b**,**e**), and recovered for 1 d after 3 d of heat treatment (**c**,**f**). Relative electrolyte leakage (**g**) and proline content (**h**) in leaves treated at 42 °C for 0 d and 3 d and recovered for 1 d. Three independent biological replicates were used in each treatment, with 9 plants (6–8 true leaves) per replicate. Data are presented as mean ±SEM of three independent biological replicates. Asterisks indicate statistically significant differences between tolerant and sensitive genotypes. **, *p* < 0.01, Student's *t* test.

In this study, relative electrolyte leakage and proline content were measured in the treated plants to evaluate heat tolerance. Under normal growth conditions, there was no significant difference in the relative electrolyte leakage and proline content between

the two varieties (Figure 1g,h). However, after 3 d of heat treatment at 42 °C and 1 d recovery at 25 °C, there was a remarkable increase in relative electrolyte leakage in the two varieties, with levels being significantly lower in 17-03 than in H1023 (Figure 1g). The proline content in plants increased after HS and recovery in both varieties (Figure 1h). After 3 d of heat treatment, the proline content increased, but there was no significant difference between the two varieties. However, the proline content further increased significantly in the recovery stage, and the proline content in 17-03 was obviously higher than that in H1023. These results show that 17-03 was more heat-tolerant than H1023, as the cell membranes were protected from damage, and osmotic stress was alleviated by increasing the levels of proline, an important osmotic protectant.

### 3.2. Overview of Transcriptomic Data for 17-03 and H1023

Transcriptome sequencing yielded 815.2 M raw reads (Table S1). After filtering, 724.31 M valid reads were obtained in 18 libraries (Table S1). The average effective data obtained from each sample were 6.03 G, accounting for 88.85% of the original data (Table S1). The Q30 base percentage of each library was above 92.81% (Table S1). The results showed that the sequencing quality was reliable, and the data were suitable for the subsequent analyses. The clean reads (82.23–85.35%) were mapped to the pepper reference genome CM334 (Table S1), indicating that the data could be used for subsequent analysis.

### 3.3. Identification of Differentially Expressed Genes (DEGs)

Based on the RNA-seq experiment, 24,448 expressed genes were identified in the pepper leaves (Table S2). Among these expressed genes, 11,633 DEGs were identified in 17-03 and H1023 among the four groups (Figure 2a; Table S3). Of the 11,633 DEGs, 7327 (3435 upregulated and 3892 downregulated) and 7778 (4025 upregulated and 3753 downregulated) DEGs were identified in groups CK1_vs_T1 and CK2_vs_T2, respectively (Figure 2a,b); 5185 DEGs were common between the two groups (Figure 2a); however, 2142 and 2593 DEGs were specially differentially expressed in CK1_vs_T1 and CK2_vs_T2, respectively (Figure 2a). Approximately, 3338 (2021 upregulated and 1317 downregulated) and 4822 (2256 upregulated and 2566 downregulated) DEGs were identified in groups CK1_vs_M1 and CK2_vs_M2, respectively (Figure 2b); there were 1934 common DEGs and 1404 and 2888 DEGs specially differentially expressed DEGs in CK1_vs_M1 and CK2_vs_M2, respectively (Figure 2a). Interestingly, 1229 common DEGs were identified among all four groups (Figure 2a).

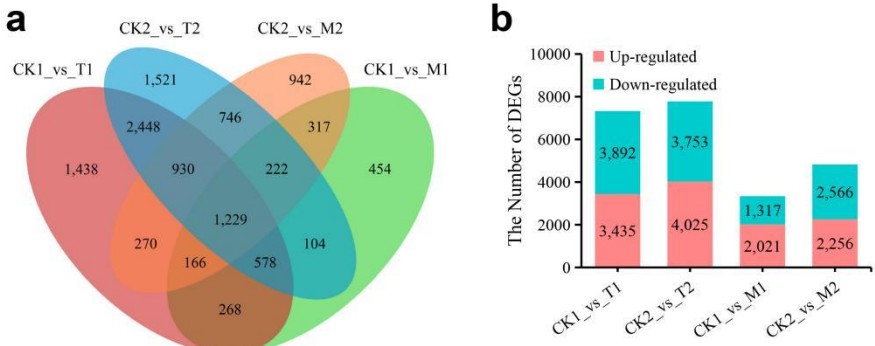

**Figure 2.** Expression analysis of differentially expressed genes (DEGs) in 17-03 and H1023 leaves after 3 d heat treatment at 42 °C and 1 d recovery at 25 °C. Numbers of DEGs in 17-03 and H1023 at different times (**a**). Numbers of up- and down regulated DEGs in 17-03 and H1023 at different times (**b**).

### 3.4. Validation of RNA-Seq Data Using qRT-PCR

To confirm the accuracy of the RNA-seq data, transcriptional levels of 14 randomly selected DEGs, representing a wide range of expression levels and patterns, were detected using qRT-PCR. All 14 DEGs participated in the process of HS response, includ-

ing small heat shock protein (CA03g21390), HS transcription factor (CA03g16300), bZIP (CA08g12820), MYB (CA04g16680, CA06g27890), WRKY genes (CA03g32070, CA08g08240, and CA09g11940), NAC genes (CA05g04410, CA07g18020, CA09g12970, and CA11g04440), EG45-like domain-containing protein (CA07g00930), and universal stress protein A-like protein (CA11g00890) (Figure 3). The fold changes varied in the RNA-Seq and qRT-PCR analyses. Generally, the expression patterns determined using qRT-PCR were consistent with those obtained using RNA-Seq (Figure 3), which confirmed the accuracy of the RNA-Seq results reported in this study.

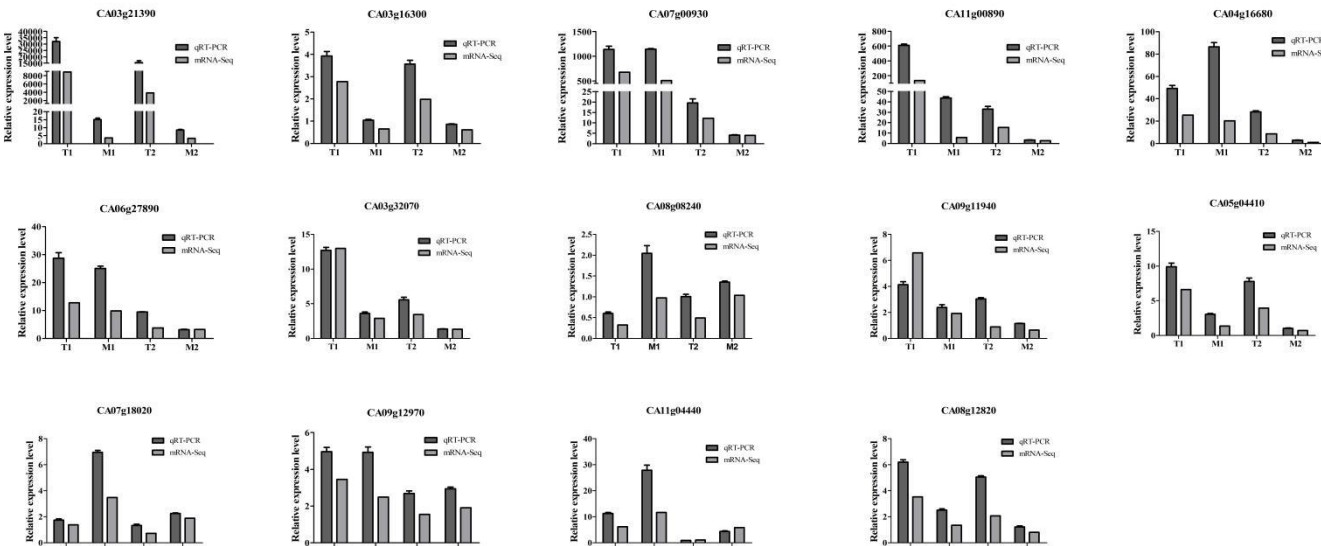

**Figure 3.** Quantitative reverse-transcription polymerase chain reaction (qRT-PCR)-based validation of differentially expressed genes (DEGs) in response to heat stress (HS) at different time intervals. Ordinate represents fold changes of RNA-Seq data and the relative expression level of qRT-PCR. The relative expression level of each gene under stress at each time point was compared with that under normal conditions. qRT-PCR data are presented as mean ± SEM of three independent technical replicates.

### 3.5. Functional Enrichment Analysis of DEGs

To explore the biological functions of DEGs in the four groups, GO enrichment analysis was performed. In total, 5133 of 11,633 DEGs in the four comparisons were annotated with GO terms and assigned to three categories: molecular function (MF), biological process (BP), and cellular component (CC) (Figure 4; Table S3). Under HS, the GO terms "photosynthesis, light harvesting" (GO:0009765), "DNA conformation change" (GO:0071103), "nucleosome assembly" (GO:0006334), "DNA packaging" (GO:0006323), "chromatin assembly" (GO:0031497), "nucleosome organization" (GO:0034728), and "chromatin assembly or disassembly" (GO:0006333) were the most commonly enriched components in the BP category in 17-03 and H1023 (Figure 4; Table S4). The most commonly enriched components were "cell wall" (GO:0005618) and "external encapsulating structure" (GO:0030312) in the CC category and "nucleosome binding" (GO:0031491) in the MF category in 17-03 and H1023 (Figure 4; Table S4). However, "DNA replication" (GO:0006260), "DNA replication initiation" (GO:0006270), "protein folding" (GO:0006457), and "protein-DNA complex assembly" (GO:0065004) in the BP category, "protein-DNA complex" (GO:0032993) in the CC category, and "nucleosomal DNA binding" (GO:0031492) in the MF category were only enriched in 17-03 (Table S4). For the recovery stage after HS, the most significantly enriched GO terms differed between 17-03 and H1023. In 17-03, DEGs were most enriched in "translation" (GO:0006412), "peptide biosynthetic process" (GO:0043043), "peptide metabolic process" (GO:0006518), "amide biosynthetic process" (GO:0043604), "cellular amide metabolic process" (GO:0043603), and "organonitrogen compound biosynthetic process" (GO:1901566) in the BP category (Table S4). "Ribosome" (GO:0005840), "non-

membrane-bounded organelle" (GO:0043228), and "intracellular non-membrane-bounded organelle" (GO:0043232) were the top three CC categories (Table S4). The terms of the MF category were "structural constituent of ribosome" (GO:0003735) and "structural molecule activity" (GO:0005198) (Table S4).

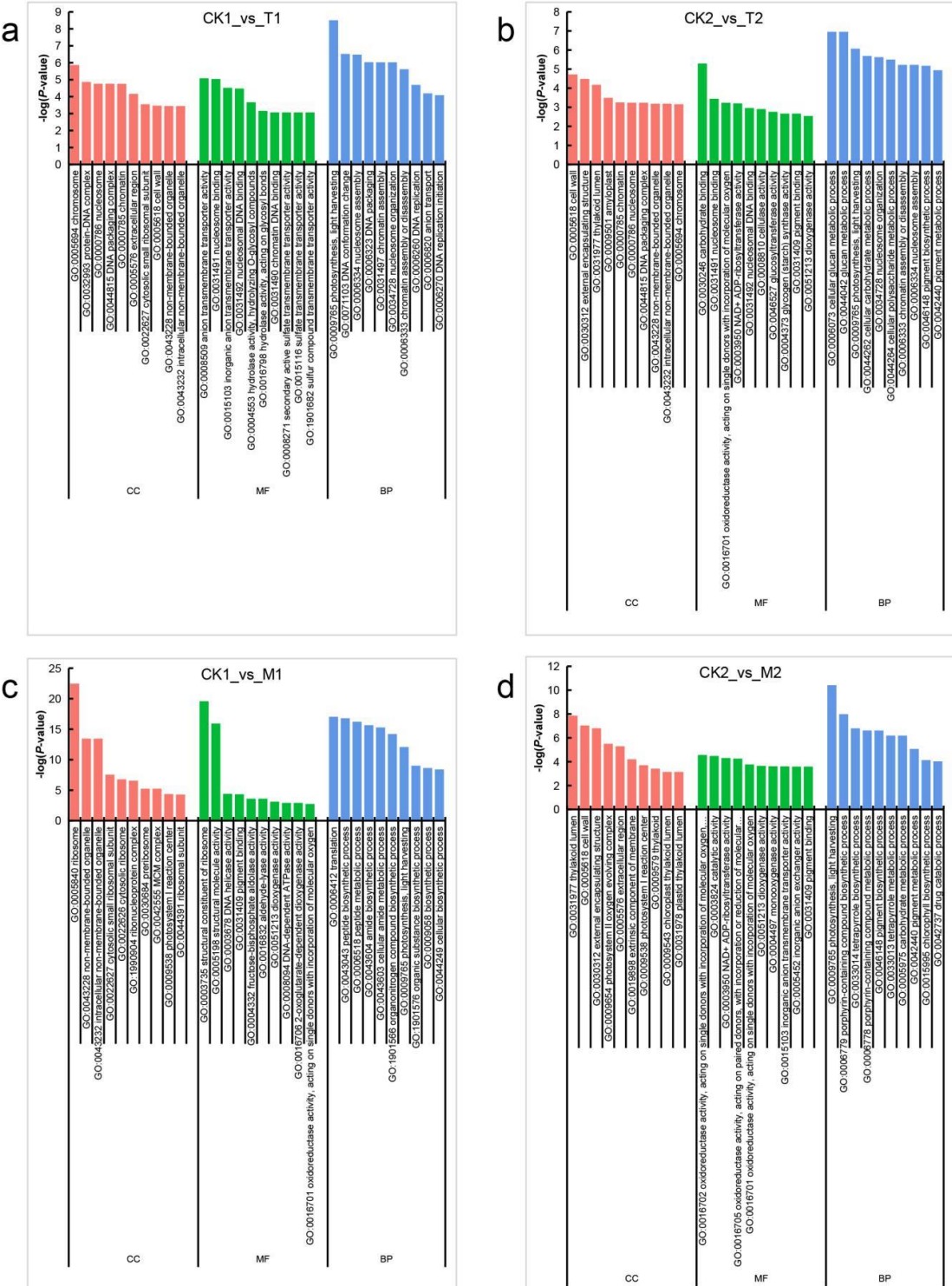

**Figure 4.** Gene ontology (GO) classifications of differentially expressed genes (DEGs) in the comparison groups CK1_vs_T1 (**a**), CK2_vs_T2 (**b**), CK1_vs_M1 (**c**), and CK2_vs_M2 (**d**). The DEGs were assigned to three categories: biological process (BP), cellular component (CC), and molecular function (MF). The *X*-axis indicates the top ten most significantly enriched BP, CC, and MF categories. The *Y*-axis indicates -log10 (*p*-value).

We performed KEGG pathway analysis to examine the pathways in which DEGs were involved. The significantly enriched pathways involved in HS and recovery responses are shown in Figure 5 and Table S5. The common significantly enriched pathways under HS were identified, such as "photosynthesis–antenna proteins" (cann00196), "ribosome biogenesis in eukaryotes" (cann03008), "fatty acid elongation" (cann03008), "anthocyanin biosynthesis" (cann00942), and "glycine, serine and threonine metabolism" (cann00260) (Figure 5 and Table S5). However, the significantly enriched KEGG pathways during recovery were different. The term "ribosome" (cann03010) was the most significantly enriched in 17-03, whereas "photosynthesis–antenna proteins" (cann00196), "carbon fixation in photosynthetic organisms" (cann00710), "porphyrin and chlorophyll metabolism" (cann00860), and "carotenoid biosynthesis" (cann00906) were the most significantly enriched in H1023 (Figure 5 and Table S5).

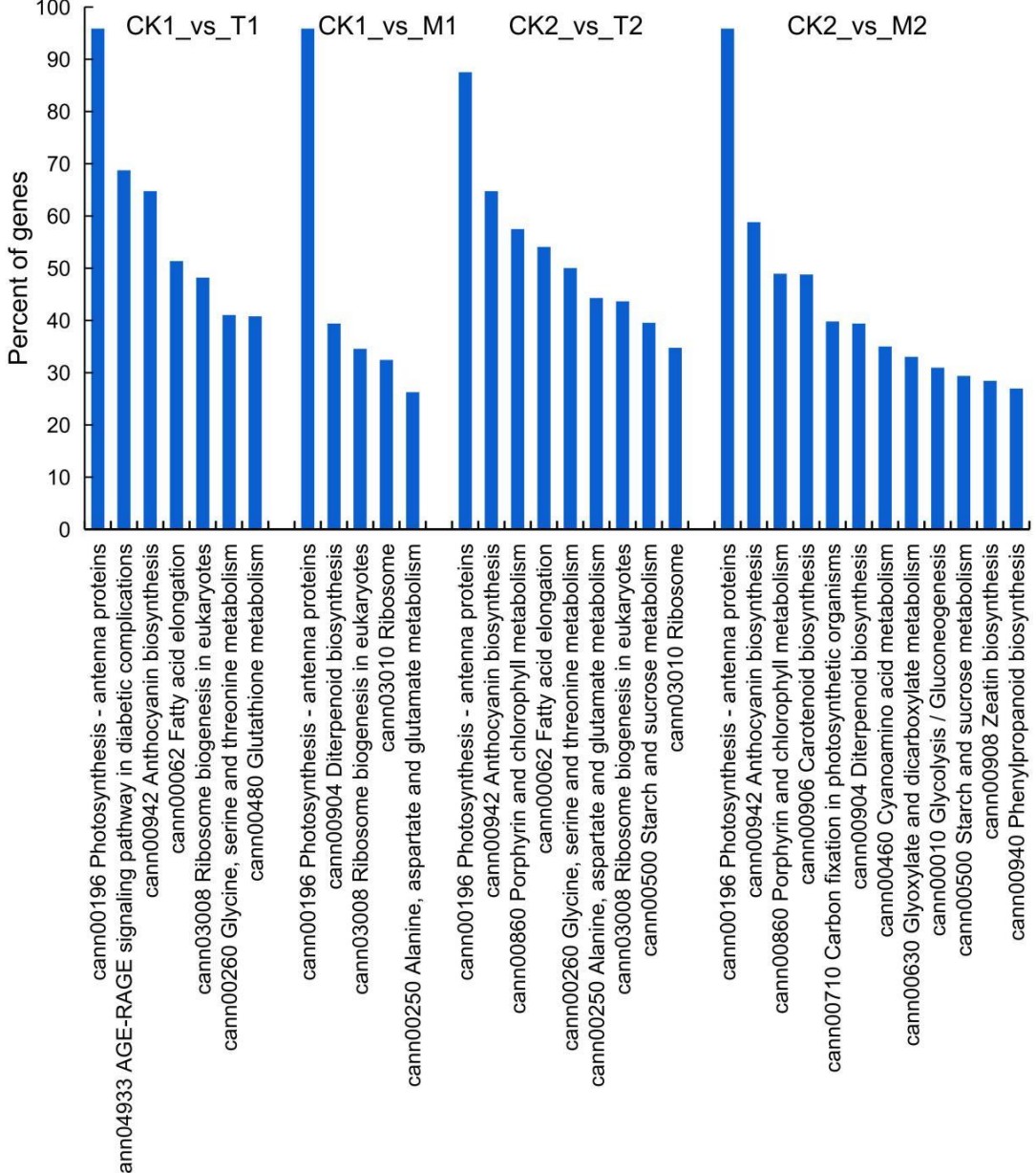

**Figure 5.** The significantly enriched pathways in comparison groups CK1_vs_T1, CK2_vs_T2, CK1_vs_M1, and CK2_vs_M2. The *X*-axis indicates the significantly enriched Kyoto Encyclopedia of Genes and Genomes (KEGG) pathways, and the *Y*-axis indicates the percentage of a specific category of genes in the main category.

### 3.6. Analysis of HS-Responsive Heat Shock Proteins and Heat Transcription Factors

Hsps and Hsfs are sensitive to HS, indicating that they play important roles in the HS response. Based on the results of transcriptome sequencing, 47 Hsp members were identified as differentially expressed in at least one of the four comparison groups (Figure 6; Table S6). Most of these were dramatically upregulated in pepper leaves after 3 d of heat treatment and were more highly expressed in H1023 than in 17-03 (Figure 6). Moreover, seven Hsps were expressed at higher levels after recovery from HS. CA01g13220, CA02g11030, CA09g06120, and CA10g10840 were highly expressed in H1023 cells (Figure 6), and CA01g31330 was highly expressed in 17-03 and H1023 cells (Figure 6). Furthermore, the expression levels of five Hsps (CA04g02800, CA09g03220, CA09g06120, CA11g13160, and CA11g13170) markedly decreased after 3 d of heat treatment (Figure 6). Similarly, 17 significantly differentially expressed Hsfs were identified, most of which were upregulated during and after recovery from HS (Figure 6). Among them, six (CA02g11030, CA03g06850, CA05g00840, CA06g08710, CA07g15920, and CA10g20440) were significantly highly expressed in H1023 after recovery from HS (Figure 6). These results indicate that these significantly differentially expressed Hsps and Hsfs might play an important role in plant protection in the long-term HS response of pepper.

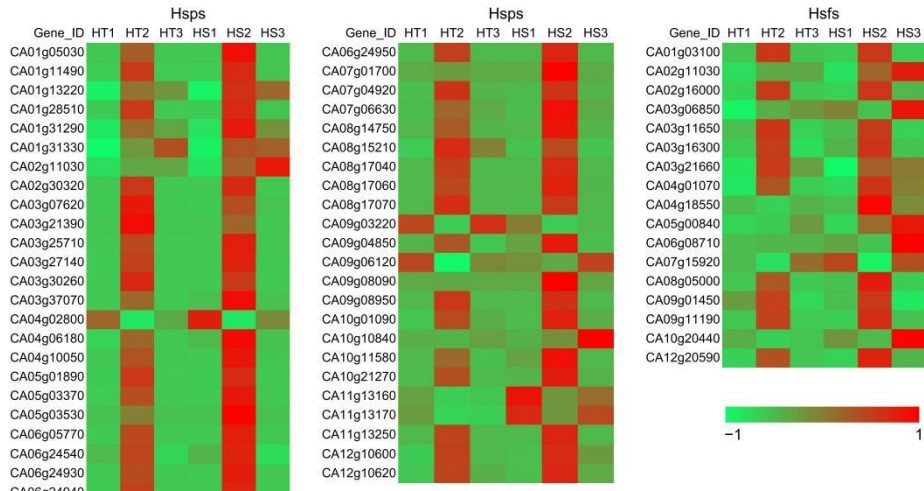

**Figure 6.** Heatmap of differentially expressed genes (DEGs) encoding Hsps and Hsfs in the comparison groups CK1_vs_T1, CK2_vs_T2, CK1_vs_M1, and CK2_vs_M2. The color gradient represents the normalized fragments per kilobase of transcript per million mapped reads (FPKM) value (Z-score) of DEGs. The redder the bars, the higher the gene expression level.

### 3.7. Analysis of HS-Responsive Transcription Factors

Transcription factors (TFs) play an important role in plant growth and development, as well as in biotic and abiotic stress response networks. Transcriptome analysis showed that many TFs in pepper were regulated by high temperatures and participated in plant recovery. A total of 49 TF families of 635 TFs were differentially expressed during heat treatment and recovery in 17-03 and H1023, including HSF, NAC, WRKY, ERF, bHLH, MYB, C2H2, B3, GRAS, bZIP, and HD-ZIP (Table S7). In this study, 38 DEGs encoding NAC proteins were identified. Among them, most were upregulated during and after recovery from HS. Moreover, the expression levels of most NAC TFs were higher in H1023 than in 17-03 during and after recovery from HS (Figure 7). WRKY proteins have also been reported to play important roles in heat response. Here, 35 DEGs encoding WRKY proteins were identified, with almost half of them upregulated in both H1023 and 17-03 during and after recovery from HS (Figure 7). Moreover, five WRKY genes (CA01g01280, CA01g34460, CA09g05110, CA09g11940, and CA12g09290) were upregulated in 17-03 but downregulated in H1023 after recovery from HS (Figure 7). Furthermore, some WRKY genes, such as CA11g05370, CA02g18540, CA09g08120, CA11g03750, and CA01g22410,

were significantly highly expressed in H1023 after recovery from HS, but not at other times in H1023 or in 17-03 (Figure 7).

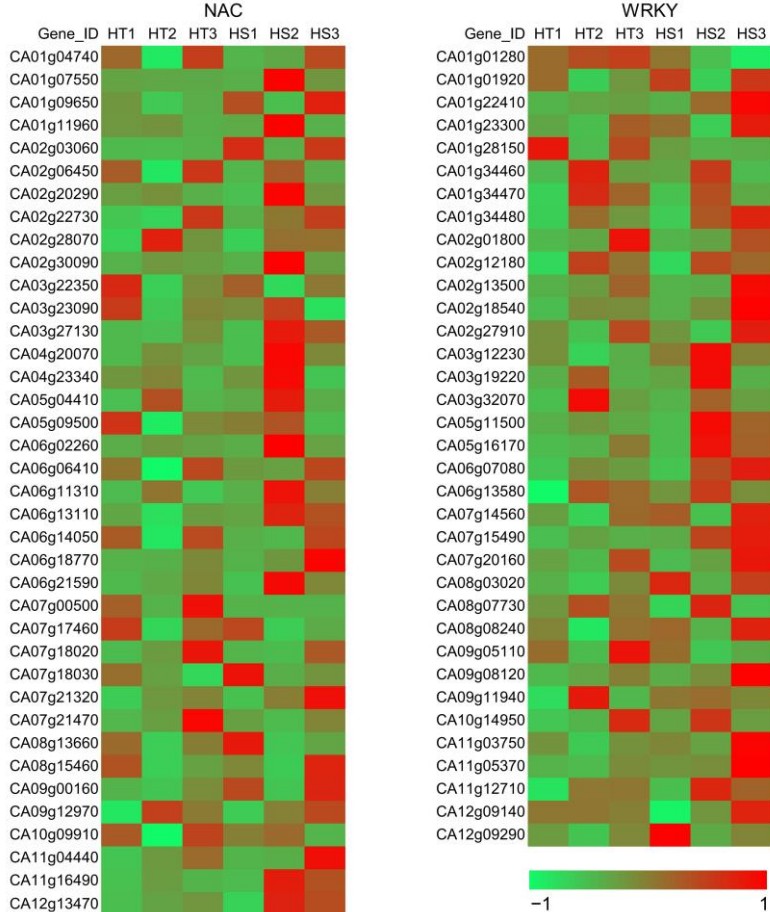

**Figure 7.** Heatmap of differentially expressed genes (DEGs) encoding NAC and WRKY proteins in the comparison groups CK1_vs_T1, CK2_vs_T2, CK1_vs_M1, and CK2_vs_M2. The color gradient represents the normalized fragments per kilobase of transcript per million mapped reads (FPKM) value (Z-score) of DEGs; the redder the bars, the higher the gene expression level.

## 4. Discussion

High temperature is one of the key climatic parameters affecting plant growth and development, resulting in crop yield losses [47]. HS can restrict photosynthesis, increase photorespiration and transpiration rate through stomatal regulation, and reduce plant biomass [7]. Pepper is a highly temperature-sensitive crop [3]. Although the physiological effects of HS on pepper have been widely studied, changes in pepper at the molecular level in response to HS and subsequent recovery are poorly understood. Therefore, to better understand the HS response in pepper, it is necessary to uncover the mechanisms underlying it. In the present study, we investigated the phenotypic and physiological changes in pepper seedlings of two varieties during and after recovery from HS. Furthermore, we comparatively analyzed heat-induced transcriptomic changes to obtain a global view of HS responses in pepper.

Under high temperature stress, the structure and function of the cell protoplasmic membrane are initially damaged, resulting in an increase in cell membrane permeability and intracellular electrolyte leakage and finally leading to an increase in electrolyte leakage of tissue leachate [48]. Therefore, the degree of electrolyte extravasation and high temperature injury can be determined by measuring the relative electrical conductivity of the tissue extract. In this study, 17-03 and H1023 were verified as high temperature-resistant and high

temperature-sensitive varieties, respectively (Figure 1b,c,f,g), and were used to explore the responses and recovery patterns of pepper to HS and the possible mechanisms of the different heat resistances. There was a remarkable increase in relative electrolyte leakage in both 17-03 and H1023 after HS, but the levels were significantly lower in 17-03 than in H1023 (Figure 1h), indicating that 17-03 could alleviate damage to cellular membranes during HS. However, there was little change in the two varieties from HS to recovery (Figure 1h), indicating that the damage to the cell membrane caused by HS is irreversible. Proline, an amino acid and a compatible solute, has been widely reported to accumulate in response to various abiotic stresses, such as high temperatures [37]. After 3 d of heat treatment and 1 d of recovery, proline levels were significantly increased in both 17-03 and H1023. While there was no significant difference in proline content between 17-03 and H1023 during heat treatment (Figure 1h), the proline content increase in 17-03 during recovery was higher than that in H1023 (Figure 1h), indicating that the self-repairing ability of 17-03 after HS was stronger than that of H1023. Based on these data, we conclude that 17-03 is more heat-tolerant, as it protects cell membranes from damage and alleviates osmotic stress by increasing the proline levels.

Moreover, we obtained accurate data from transcriptome analyses based on RNA-seq and analyzed the genes of metabolic pathways that were significantly affected by HS and participated in the process of plant restoration. In 17-03 and H1023, there were more DEGs in the HS stage than in the recovery stage (Figure 2a,b), indicating that the regulatory mechanism of HS response was more active at a transcriptional level. There were significantly more upregulated DEGs than downregulated DEGs in CK1_vs_M1 (Figure 2b). However, in the other groups, the number of up- and downregulated DEGs was almost the same (Figure 2b). Moreover, the DEGs after heat treatment were mostly different from those during recovery in 17-03 and H1023 (Figure 2a). These results indicate that the defense and recovery mechanisms of pepper may have common regulatory pathways, and that there are different pathways for response, resistance, and repair.

Out of a total of 11,633 DEGs (Table S3), 5133 were assigned a GO classification (Table S3). GO enrichment analysis showed that DEGs after heat treatment were commonly enriched in "photosynthesis, light harvesting" (GO:0009765), "cellular glucan metabolic process" (GO:0006073), and "nucleosome assembly" (GO:0006334) in 17-03 and H1023 (Table S4). These findings are similar to the HS response in sweet maize (*Zea mays* L.) [49]. DEGs were also significantly enriched in "DNA replication" (GO:0006260), "DNA replication initiation" (GO:0006270), "protein folding" (GO:0006457), "protein-DNA complex assembly" (GO:0065004), "protein-DNA complex" (GO:0032993), and "nucleosomal DNA binding" (GO:0031492) in 17-03 after heat treatment (Figure 4; Table S4), which may confer increased resistance to high temperatures. The DEGs of H1023 and 17-03 during recovery were enriched with different GO terms and KEGG pathways (Figures 4 and 5, Tables S4 and S5), indicating that the repair pathways were different, which is likely due to the different degrees of high-temperature damage. In the KEGG pathway analysis, DEGs involved in HS response were predicted to function in metabolic pathways and the biosynthesis of secondary metabolites in 17-03 and H1023, which is similar to the results of previous studies [50,51].

Hsps, which are involved in multiple biological processes, such as signal transduction during HS, and have deduced functions, such as being chaperones, the folding and unfolding of cellular proteins, and the protection of functional sites from the adverse effects of high temperature, range in molecular mass from 10 to 200 kDa [52]. Hsps have functions as molecular chaperones that affect protein quality and were initially identified as proteins that were upregulated during heat treatment [51]. Many Hsps have been detected as heat response factors in tomato [53] and grape [54] plants. In this study, a total of 47 Hsps were significantly differentially expressed in the four groups. Among them, 45 Hsps were commonly differentially expressed in 17-03 and H1023 after heat treatment (Figure 6; Table S6). Hsps can accumulate rapidly in sensitive organs and play important roles in protecting the metabolic apparatus of cells, thus acting as a key factor in the adaptation of plants to high

temperatures [55]. In this study, almost all DEGs encoding Hsps were upregulated in the four groups, the expression of which was the highest after heat treatment (Figure 6), which may play a role in protection under and after HS in 17-03 and H1023.

In addition to Hsps, various other TF genes, such as genes of Hsfs, NAC, and WRKY TFs, were also affected by HS [56]. Hsfs combine with cis-acting Hsps to play important roles in both basal and acquired thermotolerance [26,51]. Here, 17 DEGs encoding Hsfs were identified and most were upregulated in the four groups (Figure 6; Table S6). Moreover, some Hsfs were significantly highly expressed in H1023 after recovery from HS; such as CA02g11030, CA03g06850, CA05g00840, CA06g08710, CA07g15920, and CA10g20440 (Figure 6). These significantly expressed Hsfs could play important roles in the long-term HS response of pepper, by combining with the cis-acting regulatory elements, called heat shock elements, in the promoter regions of Hsps.

Plant NAC TFs have been reported to play an important role in modulating HS responses. For example, overexpression of Arabidopsis ANAC042 leads to significant thermotolerance in transgenic plants [57]. Our data indicated that the expression of 38 TFs encoded by NAC domain-containing genes was also heat-regulated (Figure 7; Table S7). Interestingly, most upregulated NAC TFs were more highly expressed in H1023 than in 17-03, during and after recovery from HS (Figure 7). WRKY TFs are one of the largest TF families in plants and have also been reported to participate in regulating plant HS response [58]. In this study, 35 WRKY TFs responded to HS in the four groups, and almost half of them positively regulated thermotolerance (Figure 7). Moreover, some WRKY TFs negatively regulated thermotolerance, such as CA01g01920 and CA01g23300 (Figure 7).

## 5. Conclusions

We verified the pepper varieties 17-03 and H1023 as being heat-resistant and heat-sensitive varieties and used RNA-seq to elucidate the effects of HS and the subsequent recovery on the expression of genes regulating the HS response and thermotolerance mechanisms. A total of 11,633 DEGs were identified in the four groups, with 1229 common DEGs among all four groups. Functional enrichment analysis showed that in 17-03 and H1023, DEGs were most enriched in metabolic processes under stress and photosynthesis and light harvesting during HS and after recovery from HS. The most significantly enriched pathways in 17-03 and H1023 were the same under HS, but differed during recovery. Furthermore, 38 Hsps, 17 Hsfs, 38 NAC TFs, and 35 WRKY TFs were identified as participating in the HS or recovery responses. These findings facilitate a better understanding of the molecular mechanisms underlying HS and recovery after HS in different pepper genotypes.

**Supplementary Materials:** The following are available online at https://www.mdpi.com/article/10.3390/horticulturae7100339/s1, Table S1: Statistical analysis of pepper clean reads in 18 libraries for RNA-seq. Table S2: Gene read count, FPKM value, annotation, and functional enrichment. Table S3: Detailed list of DEGs in 17-03 and H1023 under HS (42 °C) for 3 d and recovery (25 °C) for 1 d relative to the control. Table S4: Significantly enriched GO terms of DEGs in 17-03 and H1023 in the four groups (FDR ≤ 0.05). Table S5: Significantly enriched KEGG pathway in 17-03 and H1023 under HS (42 °C) for 3 d and recovery (25 °C) for 1 d (FDR ≤ 0.05). Table S6: DEGs encoding Hsps in 17-03 and H1023 at the heat treatment and recovery stages. Table S7: Differentially expressed transcription factors in 17-03 and H1023 at the heat treatment and recovery stages. Table S8: qRT-PCR primers used for validation of RNA-Seq data.

**Author Contributions:** Conceptualization, M.Y. and C.J.; methodology, M.Y. and C.J.; validation, F.W., Y.Y., C.Y., N.L., S.S., Y.L., S.G., C.J. and M.Y.; formal analysis, F.W., Y.Y., C.Y., N.L. and S.G.; investigation, S.S. and Y.L.; resources, F.W. and M.Y.; data curation, C.J. and M.Y.; writing—original draft preparation, F.W. and Y.Y.; writing—review and editing, C.J. and M.Y.; visualization, C.J. and M.Y.; supervision, C.J. and M.Y.; project administration, C.J. and M.Y.; funding acquisition, M.Y. All authors have read and agreed to the published version of the manuscript.

**Funding:** This work was supported by grants from the Natural Science Foundation of Hubei Province, China (2020CFA010), the earmarked fund for the China Agriculture Research System (CARS-23-G28), Key R & D Program Projects in Hubei Province (2020BBA037), China Postdoctoral Science Foundation (2017M620305), and Youth Fund of Hubei Academy of Agricultural Sciences (2021NKYJJ04).

**Institutional Review Board Statement:** Not applicable.

**Informed Consent Statement:** Not applicable.

**Data Availability Statement:** The data used for the analysis in this study are available in the article and Supplementary Materials.

**Acknowledgments:** We thank Personal Biotechnology Co., Ltd., Shanghai, China for RNA-Sequencing.

**Conflicts of Interest:** The authors declare no conflict of interest.

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
