# Peer review of "Transcriptomics Analysis of Heat Stress-Induced Genes in Pepper (Capsicum annuum L.) Seedlings"

_horticulturae, doi:10.3390/horticulturae7100339_

Round 1
Reviewer 1 Report
This manuscript presents a Transcriptomics Analysis of Heat Stress-Induced Genes in Pepper (Capsicum annuum L.) Seedlings. It is clear that the authors have done a lot of work, the structure and the use of English of the manuscript medium level of this work. The methods and results need to be revisited for completeness and accuracy. Moreover, the analysis is statistically valid. My comments include
Line 14. In this study, 17-03 and H1023 were identified as heat-tolerant and heat-sensitive varieties, respectively. Their leaves' transcript abundance was quantified using RNA-sequencing to elucidate the effect of HS and subsequent recovery on the expression of genes.
Line 29 Pepper (Capsicum annuum L.) is an important member of the Solanaceae family and one of many countries' most important spice and vegetable crops.
Line 53 In addition, HS will accumulate reactive oxygen species in plants, resulting in oxidative damage to cells. High temperature causes metabolic imbalance and production of reactive oxygen species in plants, which aggravates lipid peroxi dation and protein denaturation of cell membrane, thus affecting the structure and function of biofilm and other macromolecules severe cases leads to cell damage and plant death [10].
Line 94 is not true are
Line 140 Relative electrolyte leakage, which measures cellular membrane integrity, is frequently used to evaluate plant stress tolerance
Line 146 The solution's electrolyte leakage (R1) was measured with a portable magnetic con ductility meter (DDB-303A, Shanghai, China) at 25℃.
Line 160 ACCORDING TO THE MANUFACTURER'S INSTRUCTIONS, total RNA was extracted from 18 pepper leaf samples using TRIzol reagent (Life Technologies, California, USA).
Line 167 Then, the cDNA libraries' quality for sequencing was checked using the Agilent 2100 bioanalyzer system (Agilent Technologies, Inc., Santa Clara, CA, USA).
Line 214 were seriously wilted. In addition, the….
Line 218 did not stretch completely. In addition, the edges of ….
Line 320 We also performed the KEGG pathway analysis according to the database to examine the pathways that the DEGs are involved in.
Line 352 These results indicated that these significantly expressed Hsps and Hsfs might play an important role in plant protection in pepper's long-term heat stress response.
Line 385 crop yield loss
Line 411 There was no significant difference in proline content during the heat treatment between 17-03 and H1023.
Line 467 plant protection in pepper's long-term heat stress
Line 471 heatstress responses
Line 475 WRKY TFs were one of the largest TF families in plants, which have also been reported to participate in regulating the plant HS response [59].
good luck
Author Response
Comments and Suggestions for Authors
This manuscript presents a Transcriptomics Analysis of Heat Stress-Induced Genes in Pepper (Capsicum annuum L.) Seedlings. It is clear that the authors have done a lot of work, the structure and the use of English of the manuscript medium level of this work. The methods and results need to be revisited for completeness and accuracy. Moreover, the analysis is statistically valid. My comments include
Response: We appreciate the reviewer’s affirmation on our work. Especially, we appreciate the time the reviewer spent making these constructive comments on the manuscript. These comments are all valuable and helpful for revising and improving our manuscript.
Line 14. In this study, 17-03 and H1023 were identified as heat-tolerant and heat-sensitive varieties, respectively. Their leaves' transcript abundance was quantified using RNA-sequencing to elucidate the effect of HS and subsequent recovery on the expression of genes.
Response: It has been revised.
Line 29 Pepper (Capsicum annuum L.) is an important member of the Solanaceae family and one of many countries' most important spice and vegetable crops.
Response: It has been revised.
Line 53 In addition, HS will accumulate reactive oxygen species in plants, resulting in oxidative damage to cells. High temperature causes metabolic imbalance and production of reactive oxygen species in plants, which aggravates lipid peroxi dation and protein denaturation of cell membrane, thus affecting the structure and function of biofilm and other macromolecules severe cases leads to cell damage and plant death [10].
Response: It has been revised.
Line 94 is not true are
Response: It has been revised.
Line 140 Relative electrolyte leakage, which measures cellular membrane integrity, is frequently used to evaluate plant stress tolerance
Response: It has been revised.
Line 146 The solution's electrolyte leakage (R1) was measured with a portable magnetic con ductility meter (DDB-303A, Shanghai, China) at 25℃.
Response: It has been revised.
Line 160 ACCORDING TO THE MANUFACTURER'S INSTRUCTIONS, total RNA was extracted from 18 pepper leaf samples using TRIzol reagent (Life Technologies, California, USA).
Response: It has been revised.
Line 167 Then, the cDNA libraries' quality for sequencing was checked using the Agilent 2100 bioanalyzer system (Agilent Technologies, Inc., Santa Clara, CA, USA).
Response: It has been revised.
Line 214 were seriously wilted. In addition, the….
Response: It has been revised.
Line 218 did not stretch completely. In addition, the edges of ….
Response: It has been revised.
Line 320 We also performed the KEGG pathway analysis according to the database to examine the pathways that the DEGs are involved in.
Response: It has been revised.
Line 352 These results indicated that these significantly expressed Hsps and Hsfs might play an important role in plant protection in pepper's long-term heat stress response.
Response: It has been revised.
Line 385 crop yield loss
Response: It has been revised.
Line 411 There was no significant difference in proline content during the heat treatment between 17-03 and H1023.
Response: It has been revised.
Line 467 plant protection in pepper's long-term heat stress
Response: It has been revised.
Line 471 heatstress responses
Response: It has been revised.
Line 475 WRKY TFs were one of the largest TF families in plants, which have also been reported to participate in regulating the plant HS response [59].
Response: It has been revised.
Reviewer 2 Report
The study by Wang et al. focused on implementing an RNA-sequencing approach to dissect the genetic architecture of heat stress-induced genes in chile pepper seedlings. Overall, the manuscript was well-written and can be accepted in its current form.
Minor comment: There should be consistency on the formatting for gene names, i.e., they should be italicized throughout the manuscript.
Author Response
Comments and Suggestions for Authors
The study by Wang et al. focused on implementing an RNA-sequencing approach to dissect the genetic architecture of heat stress-induced genes in chile pepper seedlings. Overall, the manuscript was well-written and can be accepted in its current form.
Minor comment: There should be consistency on the formatting for gene names, i.e., they should be italicized throughout the manuscript.
Response: We appreciate the reviewer’s affirmation on our work. Especially, we appreciate the time the reviewer spent making these constructive comments on the manuscript. It has been revised that genes were italicized throughout the manuscript.
Reviewer 3 Report
line 28, it is not explained what it means here (when it first appears in the text), NAC proteins, 35 WRKY.
likewise, in line 28 (ATP), 51 ROS, 87 (BR).
line 68 instead of "hormones, etc.", is correct hormones etc.,
Author Response
Comments and Suggestions for Authors
line 28, it is not explained what it means here (when it first appears in the text), NAC proteins, 35 WRKY.
likewise, in line 28 (ATP), 51 ROS, 87 (BR).
Response: It has been revised as “NAC (NAM, ATAF1/2, CUC2) proteins and 35 WRKY ” in line 28, “adenosine triphosphate (ATP) synthesis and carbon fixation” in line 50, “reactive oxygen species (ROS) accumulates greatly,” in line 51 and “BZR1, the key regulator of brassinoid (BR) response, regulates HS response of tomato through RBOH1-dependent ROS signal, at least in part through the regulation of FER2 and FER3 to achieve” in line 87.
line 68 instead of "hormones, etc.", is correct hormones etc.,
Response: It has been revised.